# A simple model for glioma grading based on texture analysis applied to conventional brain MRI

**José Gerardo Suárez-García**, **Javier Miguel Hernández-López**, **Eduardo Moreno-Barbosa**, **Benito de Celis-Alonso***

Faculty of Physics and Mathematics, Benemérita Universidad Autónoma de Puebla (BUAP), Puebla, Puebla, México

* bdca_BUAP@yahoo.com.mx

**Data Availability Statement:** All MRI Images and segmentation files are available from http://www.med.upenn.edu/sbia/brats2018/registration.html.

## Abstract

Accuracy of glioma grading is fundamental for the diagnosis, treatment planning and prognosis of patients. The purpose of this work was to develop a low-cost and easy-to-implement classification model which distinguishes low-grade gliomas (LGGs) from high-grade gliomas (HGGs), through texture analysis applied to conventional brain MRI. Different combinations of MRI contrasts ($T_{1Gd}$ and $T_2$) and one segmented glioma region (necrotic and non-enhancing tumor core, NCR/NET) were studied. Texture features obtained from the gray level size zone matrix (GLSZM) were calculated. An under-sampling method was proposed to divide the data into different training subsets and subsequently extract complementary information for the creation of distinct classification models. The sensitivity, specificity and accuracy of the models were calculated, and the best model explicitly reported. The best model included only three texture features and reached a sensitivity, specificity and accuracy of 94.12%, 88.24% and 91.18%, respectively. According to the features of the model, when the NCR/NET region was studied, HGGs had a more heterogeneous texture than LGGs in the $T_{1Gd}$ images, and LGGs had a more heterogeneous texture than HGGs in the $T_2$ images. These novel results partially contrast with results from the literature. The best model proved to be useful for the classification of gliomas. Complementary results showed that the heterogeneity of gliomas depended on the MRI contrast studied. The chosen model stands out as a simple, low-cost, easy-to-implement, reproducible and highly accurate glioma classifier. Importantly, it should be accessible to populations with reduced economic and scientific resources.

## Introduction

Gliomas are tumors formed by the glial cells of nervous tissue. These can be benign or malignant. Malignant gliomas represent about 80% of all malignant brain tumors [1], and can be classified as low-grade gliomas (LGGs; grade II, according to the World Health Organization, WHO) or high-grade gliomas (HGGs; grades III and IV, according to the WHO) [2]. LGGs

All other relevant data are within the paper and its Supporting Information files.

**Funding:** The author JGSG was supported by the National Council of Science and Technology (https://www.conacyt.gob.mx/), through a scholarship for postgraduate studies (grant number 461568). The funders had no role in study design, data collection and analysis, decision to publish, or preparation of the manuscript.

**Competing interests:** The authors have declared that no competing interests exist.

are usually slow-growing and infiltrative tumors. Treatment consists of a complete resection of the tumor and subsequent follow-up with the patient. In some cases, chemotherapy and radiotherapy may be necessary. On the other hand, HGGs evolve rapidly and immediate treatment is necessary, including complete resection, chemotherapy and radiotherapy [3]. HGG patients have a low life expectancy (approximately 1 to 2 years), while LGG patients have a longer life expectancy (approximately 5 to 10 years) [4]. The 2016 WHO Classification of Tumors of the Central Nervous System showed that molecular characteristics are of greater importance, compared to histological characteristics, for the diagnosis and management of each patient [2]. However, as in other medical classification problems, a single approach is usually not sufficient to provide all the information necessary for the understanding of a disease and the accuracy of its diagnosis [5]. On the other hand, the latest advances in disease diagnosis are not always accessible to the entire population, as is the case in developing countries. Therefore, the creation of low-cost and relatively easy-to-implement diagnostic methodologies is useful and necessary.

Different imaging techniques, such as computed tomography (CT), positron emission tomography (PET), single-photon emission computed tomography (SPECT), magnetic resonance imaging (MRI), infrared spectroscopic imaging, or a combination of these, have been used for the study and classification of gliomas [6–14]. However, intracranial tumors are best evaluated on MRI [15] and conventional MRI is traditionally employed in many works whose objective is to distinguish LGGs from HGGs [16]. Moreover, MRI is first utilized clinically when, after a general medical examination, the presence of a brain tumor is suspected. Once its existence is confirmed, conventional and advanced MRI can be used to delimit the tumor, to follow its evolution, and to obtain some evidence about its type and malignancy grade [17]. However, the characterization of gliomas using imaging is difficult, since they can present a mixture of low-grade and high-grade characteristics. Currently, the standard diagnosis of gliomas is performed by histopathological tests after performing a surgical resection or a stereotactic biopsy (which can be guided by MRI) [18]. These procedures are invasive and can be risky due to the location of the tumor. Moreover, due to the heterogeneity of gliomas, a biopsy presents problems such as taking samples that are not representative of the complete tumor as well as variability in the interpretation [19]. Therefore, there is a need in the clinical community to develop new non-invasive and preferably automatic diagnostic methodologies, so that the diagnosis, treatment planning and prognosis of glioma patients can be improved.

To date, various computational methodologies have been developed for the classification of gliomas. Some of them study conventional (anatomical) [7, 20] or advanced MRI (perfusion or diffusion weighted imaging, spectroscopy, etc.) [5, 21–23]. They use qualitative, semiquantitative or quantitative variables, or a combination. Some variables are obtained from a specific MRI submodality (for example, diffusion variables). However, there are quantitative analytical methodologies that allow measuring variables from all MRI. One of the most-used is texture analysis [20, 23, 24], wich consists of quantifying the spatial distribution of pixels (2D images) or voxels (3D images) with different gray levels intensities, and extracting information through statistical variables (for example, correlation, homogeneity, contrast, entropy, etc.) [25]. Color images with red/green/blue format can be analyzed by separating their three components as gray level images. Thus, texture analysis is versatile for any type of MRI [26]. In this type of analysis, texture features are extracted from the calculation of texture matrices. Among these are, for example, the so-called gray level size zone matrix (GLSZM). The GLSZM measures the number of gray level zones, $i$, and their sizes, $j$ [27]. It calculates the number of times that gray level voxels $i$ were grouped to form a set (zone) of $j$ voxels, considering all possible directions (26 in 3D). Since its invention, the GLSZM has proven to be useful when the main characteristic under study is heterogeneity [27]. Since different glioma grades are characterized by having

different levels of heterogeneity [28], characterizing gliomas through texture features obtained from the GLSZM is convenient.

Simple mathematical methods (linear regression, logistic regression, etc.) and complex mathematical methods (fuzzy modeling, artificial neural networks, etc.) have been used for glioma grading [29]. Although the most complex models tend to be the most flexible, the computational cost is also higher [30]. On the other hand, the utility of any model is usually measured after validation. For classification models, this commonly consists of using training samples to create the model and testing samples to ratify its results. However, because the available databases tipically contain small samples, it is not always possible to validate the results [19, 21–23]. Moreover, many of the databases used in different studies are private, or were acquired using specific protocols, which does not always allow their results to be extrapolated or reproduced through independent studies. Also, a common problem present in many databases is the so-called "class imbalance", which occurs when one or more classes have a greater or lesser number of samples than the others. The consequence is that a classifier will be biased towards the classes with the highest number of samples [31]. There are several strategies to deal with this problem, such as under-sampling (elimination of samples from major classes), over-sampling (replication of samples from minority classes), cost-sensitive learning (taking into account misclassification costs), and others [32].

The objective of the present work was to develop a non-invasive, semi-automatic, simple and reproducible method to differentiate low-grade and high-grade gliomas, with the benefit that it can be used in developing countries with limited access to technology. This was achieved by studying different conventional MRI contrasts, applying texture analysis (which is low-cost and easy-to-implement) and finally reporting explicitly the best classification model. In addition, since an imbalanced database was studied, an under-sampling approach was proposed, in which different subsets of gliomas with balanced classes were first created, and then complementary information from each of them was extracted in order to create a variety of classification models.

## Materials and methods

### Patient database

The patient database belonging to the *Multimodal Brain Tumor Segmentation (BRATS) Challenge 2018* [33–36] was used in this work. It was available online via a data request on the challenge page [33]. The database was comprised of routine, clinically acquired 1.0 T, 1.5 T and 3.0 T pre-operative multimodal MRI scans of 210 glioblastoma (GBM/HGG) and 75 LGG patients, with pathologically confirmed diagnosis. There was information about the age and overall survival of 168 of the 210 HGG patients, who had an average age of $60.33 \pm 12.08$ years and an average overall survival of $423 \pm 349$ days. The scans consisted of four conventional MRI contrasts: native ($T_1$) and post-contrast T1-weighted ($T_{1Gd}$), T2-weighted ($T_2$), and T2 Fluid Attenuated Inversion Recovery (FLAIR). Images of the tumors were segmented and manually labeled in different regions: Gd-enhancing tumor (ET), necrotic and non-enhancing tumor core (NCR/NET) and peritumoral edema (ED). The manual segmentation was performed by one to four raters, and their annotations were approved by experienced neuroradiologists. The database was grouped into three sets, identified as BRATS 2013 (from the 2013 challenge database), with 10 LGGs and 20 HGGs; TCIA (The Cancer Imaging Archive), with 65 LGGs and 102 HGGs; and CBICA (Center for Biomedical Image Computing and Analytics), with 88 HGGs (S1 Table). However, the scans came from 19 different institutions and were acquired with different clinical protocols as well as various scanning systems.

## Pre-processing

The scans from the database were already pre-processed [34]. Each patient's image volumes were co-registered rigidly to the $T_{1Gd}$ MRI and all images were resampled to 1 mm isotropic resolution in a standardized axial orientation with a linear interpolator. A rigid registration model was used with the mutual information similarity metric, through the Insight Segmentation and Registration Toolkit (ITK) software [37] ("VersorRigid3DTransform" with "MattesMutualInformation" similarity metric and three multi-resolution levels). All images were skull-stripped.

As the features of interest in the texture analysis describe different properties, based on the gray level intensity of the images, two extra pre-processing steps were performed before the analysis: intensity inhomogeneity correction and intensity normalization. Intensity inhomogeneities are mainly produced by imperfections in the radiofrequency coils and object-dependent interactions; in the images they are observed as a low frequency variation of the intensity across the image [38]. In any quantitative image analysis, a tissue is considered to be represented by similar gray level intensities, so that intensity inhomogeneities have a large influence on the results obtained. Therefore, it was necessary to include inhomogeneity correction in the pre-processing for this work. Besides, as images were obtained through different clinical protocols and scanning systems, their intensity ranges were different. Thus, to be able to compare the images, intensity normalization was necessary.

Inhomogeneity correction was carried out using the FreeSurfer Software Suite version 6.0 in Linux (Ubuntu 14.04) [39], through the tool "nu_correct", which applies the N3 (nonparametric non-uniformity intensity normalization method) algorithm developed by the Montreal Neurological Institute (MNI). This analyzes the image intensity distribution in order to find the smooth intensity non-uniformity field that maximizes its frequency content [40]. For each patient, this pre-processing was applied to whole brain images (including the glioma region).

Intensity normalization was performed with an algorithm developed in MATLAB and available online [41, 42]; it was based on the method proposed by Nyul et al [43], in which landmarks are adjusted on different histograms. This process of normalization required a set of reference volumes, whose selection was arbitrary. The selection criterion used exclusively for this work consisted of the following: considering that the range of intensities of each patient volume was different, then those with the lowest and highest ranges were chosen. This was done by averaging the intensities of their voxels excluding the region corresponding to the tumor; since the tumor environment is highly heterogeneous, it had to be excluded from the reference volumes used for normalization. For each of the available MRI contrasts ($T_1$, $T_{1Gd}$, $T_2$ y FLAIR), glioma grades (LGG and HGG) and datasets in the database (BRATS 2013, TCIA and CBICA), two reference volumes were chosen: one with the lowest mean intensity and the another with the highest. Thus, a total of 40 reference volumes were chosen (since CBICA did not have LGGs). Then, to normalize the rest of the $T_1$ volumes, all reference $T_1$ volumes were used, and so on with the other MRI contrasts. The normalization range was selected to vary between 0 and 255 in steps of 1 (0 corresponded to the absence of a value). Gliomas whose volumes were employed for normalization were excluded from further work. Although in total there were 40 reference volumes (16 LGGs and 24 HGGs), some of them corresponded to the same gliomas (for example, for more than one MRI contrast, the same glioma had the lowest or highest average intensity). Thus, the volumes of 11 different LGGs and 19 different HGGs were used for normalization (S2 Table). In the end, 64 LGGs and 191 HGGs were available, this being a database with imbalanced classes.

## Database division

Low-grade and high-grade gliomas were divided into two classes: training gliomas and testing gliomas. As the first part of the proposed under-sampling approach, a unique and independent

subset formed by testing gliomas (testing subset) and different subsets formed by training gliomas (training subsets) were created. In each subset the same number of LGGs and HGGs was chosen. Classifiers were created from the training subsets and then these were applied to the testing subset.

To form subsets with balanced classes the following was done. From the 64 LGGs and 191 HGGs, 34 LGGs and 34 HGGs were randomly chosen to form the testing subset (S3 Table). Then, of the remaining 157 HGGs, 30 were chosen randomly, and this was repeated 100 times. Then, along with the remaining 30 LGGs (after having chosen the testing LGGs), 100 training subsets were formed (S4 Table; Fig 1), to extract different but complementary information from different training subsets, even though the training LGGs were the same in each one.

## Texture features

In this work the so-called gray level size zone matrix (GLSZM) was calculated within the tumor regions of interest, and 13 standard texture features were obtained from it [27, 44]. This was taken as a first study approach, using a small set of texture features in order to simplify the analysis and models. Besides, other texture matrices were not used, since the results obtained with the GLSZM were good. Names and notation for the 13 features are shown in Table 1; their definitions and descriptions can be found in the literature [45].

The calculation of the GLSZM and the texture features, in addition to all work that will be described below, were completed through computational algorithms developed in the lab using MATLAB version 8.5.0.197613 (R2015a), on a normal computer system (Intel Core i7-4790 CPU at 3.60 GHz, 16 GB RAM, Windows 7).

## MRI contrasts and tumor regions

For simplicity, and in order to reduce the time consumed by the computational algorithms developed, only two of the four MRI contrasts ($T_{1Gd}$ and $T_2$), and two of the three tumor regions (NCR/NET and ED), were analyzed. The reason for choosing these MRI contrasts was that, in initial versions of the work, better results were obtained when those contrasts were analyzed, compared to the rest. On the other hand, since the Gd-enhancing tumor region was not present in all LGGs, it was excluded from the work.

All possible combinations of MRI contrasts and tumor regions were studied. Thus, the total number of combinations was 15, varying from one MRI contrast and one tumor region ($MRI^{reg}$) on their own, to all of them together. In each combination, 13 texture features were calculated for each $MRI^{reg}$. Therefore, as can be seen in Table 2, for the first four combinations, 13 features were calculated; for combinations 5 to 10, 26 were calculated; for combinations 11 to 14, 39 were calculated; and for combination 15, 52 were calculated.

## Classification models

Once the data division was made, the proposed under-sampling approach continued as follows. For each of the 15 combinations, different classification models were created. In general terms, the models were constructed from the training subsets that had the same texture features with the higher significant differences in an ordered manner (according to the $p$-values obtained after applying statistical tests). Then, different models were created and averaged. Thus, unique models of classification using from one to more texture features were obtained. The procedure for the creation of models considering some particular combination is described below. This same procedure was followed for all 15 combinations.

**Features with significant differences.** In each of the 100 training subsets, the texture features (13, 26, 39 or 52, depending on the combination) of the respective 30 LGGs and 30

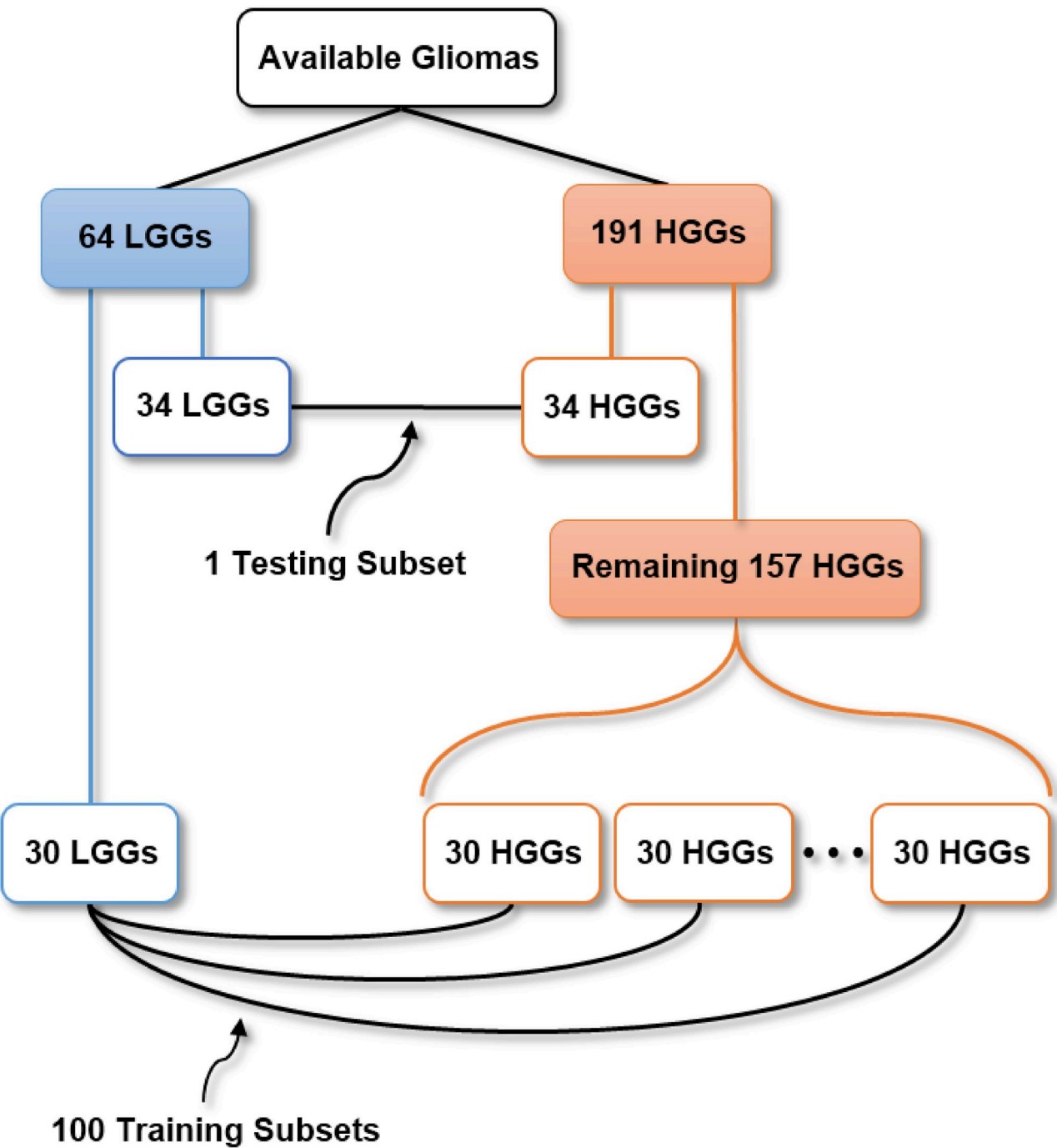

**Fig 1. Division of data and under-sampling.** From the available 64 LGGS and 191 HGGs, one testing subset and 100 training subsets with balanced classes were created by randomly choosing gliomas.

HGGs were compared. The comparison was made by applying the Wilcoxon rank-sum test. The features were ordered according to their $p$-value, putting in first place the one with the lowest $p$-value and putting in last place the one with the highest $p$-value. Then, in each subset, only the features that presented significant differences ($p < 0.05$) were considered. The number of these features was called $D_i$, with $i = 1, 2, ..., 100$. Afterward, the minimum of these

**Table 1. Studied texture features.**

| Texture feature | Notation |
|---|---|
| Small Zone Emphasis | $F_{szm.sze}$ |
| Large Zone Emphasis | $F_{szm.lze}$ |
| Gray-Level Nonuniformity | $F_{szm.glnu}$ |
| Zone-Size Nonuniformity | $F_{szm.zsnu}$ |
| Zone Percentage | $F_{szm.z.perc}$ |
| Low Gray-Level Zone Emphasis | $F_{szm.lgze}$ |
| High Gray-Level Zone Emphasis | $F_{szm.hgze}$ |
| Small Zone Low Gray-Level Emphasis | $F_{szm.szlge}$ |
| Small Zone High Gray-Level Emphasis | $F_{szm.szhge}$ |
| Large Zone Low Gray-Level Emphasis | $F_{szm.lzlge}$ |
| Large Zone High Gray-Level Emphasis | $F_{szm.lzhge}$ |
| Gray-Level Variance | $F_{szm.gl.var}$ |
| Zone-Size Variance | $F_{szm.zs.var}$ |

Names and notation for the thirteen texture features [45].

values was calculated, and called $d$ (i.e., $d = \min\{D_i\}$). Thus, a set of features $\{X_{is}\}$ was obtained, with $i = 1, 2, \ldots, 100$ (indicating the training subset) and $s = 1, 2, \ldots, d$ (indicating the order). Subsequently, $d$ histograms were created, each of them formed from the features located in the same place (from the 100 training subsets; Fig 2). That is, one histogram with all features located in the first place, another with those located in the second place, and so on until the one with those located in the position $d$. From each of these histograms, the highest frequency feature was chosen. Then, a set $\{x_s\}$, with $s = 1, 2, \ldots, d$, of ordered highest frequency features

**Table 2. Combinations studied, and numbers of calculated features.**

| Number | Combination | Total calculated features |
|---|---|---|
| 1 | $T_{1Gd}^{1}$ | 13 |
| 2 | $T_{1Gd}^{2}$ | 13 |
| 3 | $T_{2}^{1}$ | 13 |
| 4 | $T_{2}^{2}$ | 13 |
| 5 | $T_{1Gd}^{1}$-$T_{1Gd}^{2}$ | 26 |
| 6 | $T_{1Gd}^{1}$-$T_{2}^{1}$ | 26 |
| 7 | $T_{1Gd}^{1}$-$T_{2}^{2}$ | 26 |
| 8 | $T_{1Gd}^{2}$-$T_{2}^{1}$ | 26 |
| 9 | $T_{1Gd}^{2}$-$T_{2}^{2}$ | 26 |
| 10 | $T_{2}^{1}$-$T_{2}^{2}$ | 26 |
| 11 | $T_{1Gd}^{1}$-$T_{1Gd}^{2}$-$T_{2}^{1}$ | 39 |
| 12 | $T_{1Gd}^{1}$-$T_{1Gd}^{2}$-$T_{2}^{2}$ | 39 |
| 13 | $T_{1Gd}^{1}$-$T_{2}^{1}$-$T_{2}^{2}$ | 39 |
| 14 | $T_{1Gd}^{2}$-$T_{2}^{1}$-$T_{2}^{2}$ | 39 |
| 15 | $T_{1Gd}^{1}$-$T_{1Gd}^{2}$-$T_{2}^{1}$-$T_{2}^{2}$ | 52 |

Fifteen different combinations of MRI contrasts and glioma regions are listed. For MRI contrast ($T_{1Gd}$ and $T_2$), the region (reg) of the tumor studied is indicated by a superscript. Superscript 1 corresponds to the NCR/NET region, while superscript 2 corresponds to the ED region. For each $MRI^{reg}$, thirteen different texture features were calculated. The total number of calculated features in each combination is indicated.

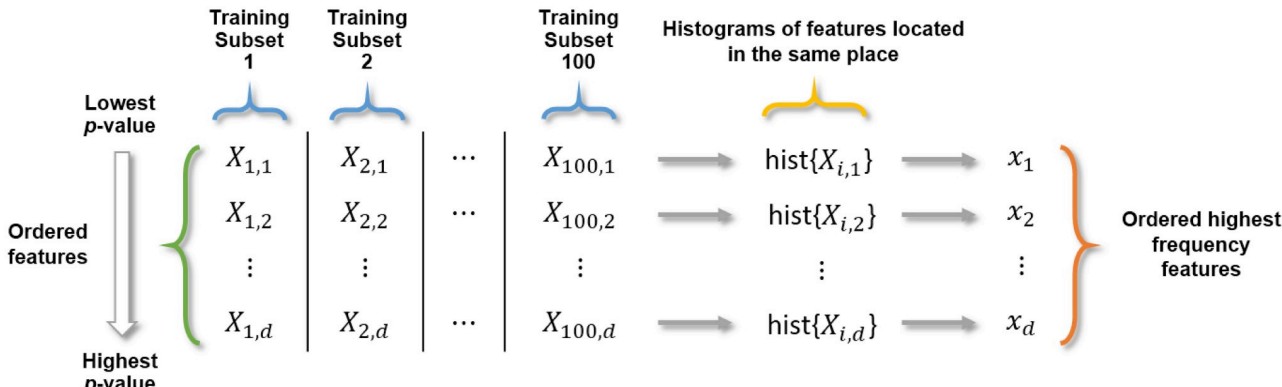

**Fig 2. Obtaining the ordered highest frequency features.** Considering the $d$ first ordered features (according to their $p$-value) of each training subset, histograms of the features located in the same place were created. Then, from the histograms, the highest frequency features were obtained.

was obtained. In the case that a histogram had the same chosen features as a previous histogram, then the next highest frequency feature was selected, so that in the end $d$ different features were obtained.

**Creation of unique classification models.** For some $t$, with $t \leq d$, training subsets whose $t$ first ordered features coincided with the $t$ first ordered highest frequency features, were chosen. The total number of subsets that complied with the above was called $w$. In each of the $w$ training subsets a multiple linear regression was carried out, employing the set of $t$ features. In the regressions, the independent variable was chosen arbitrarily to be equal to -10 for the LGGs and equal to 10 for the HGGs. Hence, $w$ individual regression models were obtained in the form:

$$\beta_{1,1}x_1 + \beta_{1,2}x_2 + \ldots + \beta_{1,t}x_t + \beta_{1,t+1} = \hat{y}_1 \beta_{2,1}x_1 + \beta_{2,2}x_2 + \ldots + \beta_{2,t}x_t + \beta_{2,t+1}$$

$$= \hat{y}_2 \vdots \beta_{w,1}x_1 + \beta_{w,2}x_2 + \ldots + \beta_{w,t}x_t + \beta_{w,t+1} = \hat{y}_w$$

in which the $\beta$'s were the coefficients obtained after performing the linear regression, $x$'s were the variables or ordered highest frequency features, and $\hat{y}$'s were predictions of the models. Then, in order to obtain a single model from the $w$ created, coefficients associated with the same variable (including the constant-term coefficient) were averaged. Thus, a unique classification model of $t$ variables was obtained and expressed as:

$$\bar{\beta}_1 x_1 + \bar{\beta}_2 x_2 + \ldots + \bar{\beta}_t x_t + \bar{\beta}_{t+1} = \hat{y} \tag{1}$$

where the $\bar{\beta}$'s were the averaged coefficients. The above was repeated for all possible values of $t$ (from 1 to $d$). Then, for the combination under consideration, $d$ different models were obtained using from 1 to $d$ variables; i.e., one model used only the ordered highest frequency feature located in the first place, another model used the two features located in the first and second places, and so on until the model using all $d$ features.

## Application of models

Unique models created from all combinations were applied to the testing subset (34 LGGs and 34 HGGs). If the prediction $\hat{y}$ of some model was less than sero, then the glioma was classified as LGG; and if it was greater than 0, then the glioma was classified as HGG. Sensitivity,

specificity, accuracy and mean absolute error (mae) were calculated for all the models. However, before calculating the mae of each model, the following was done. For any testing LGG, if its prediction $\hat{y}$ was less than -10, then this was equalized to -10, while for any testing HGG, if its prediction $\hat{y}$ was greater than 10, then it was equalized to 10. This was done because a value greater than 10 for some testing HGG, or a value less than -10 for some testing LGG, was not considered a bad result; however, the mae of the respective model would has been negatively influenced by this. Thus, there was only interest in the prediction errors of LGGs above -10 and HGGs below 10. It should be mentioned that, in the results section, the actual predictions of each glioma were shown graphically. From all unique models created, in this work only the one that obtained the best results was reported.

## Reduced models

Assuming that the best model was created from more than one variable, reduced models were created using all possible combinations of those variables. This was done with the objective of knowing if any of the variables could be left out while still obtaining good results. To understand how training subsets were chosen (among the 100 available) in order to create the reduced models, the following example can be considered. Suppose that the best model used three variables. Then, considering their order, the possible combinations of variables were 1, 2, 3, 1-2, 1-3, 2-3 and 1-2-3. For combination 1, the training subsets whose first ordered variable was variable 1 were chosen; for combinations 2 and 1-2, the subsets whose first two ordered variables were variables 1 and 2 were chosen; and for combinations 3, 1-3, 2-3 and 1-2-3, the subsets whose first three ordered variables were variables 1, 2 and 3 were chosen. Once the variables and their respective training subsets were considered, a procedure similar to that explained in the previous sections was completed, creating individual reduced models and obtaining unique reduced models. These latest models were applied to the testing subset and their sensitivity, specificity, accuracy and mae were calculated. From all unique reduced models, the one that obtained the best results with the fewest variables and lowest mae value was chosen. Its mathematical expression was then explicitly reported. Further, boxplots of the variables used in the best model and obtained from the testing subset were created, and the Wilcoxon rank-sum test was applied.

With the best classification model established, the duration of the entire classification process for each testing glioma was measured and averaged. This consisted of the following procedures: inhomogeneity correction, intensity normalization, calculation of the GLSZM, calculation of the texture features, application of the model and classification (Fig 3).

## Results

After calculating the texture features for the training HGGs and LGGs (S5, S6, S7 and S8 Tables), the minimum number of features (called $d$) with significant differences ($p < 0.05$) of two MRI$^{\text{reg}}$, $T_{1Gd}{}^{2}$ and $T_2{}^{2}$ (numbers of combinations 2 and 4, respectively, in Table 2), whose study region was ED (indicated by the superscript 2), was equal to zero. That is, when the only glioma region studied was ED, in some of the 100 training subsets there were no texture features with significant differences between LGGs and HGGs. Therefore, it was decided to exclude from the subsequent work those combinations that included the two mentioned MRI$^{\text{reg}}$. Then, of the total of 15 combinations between MRI contrasts and tumor regions, only three continued to be studied: $T_{1Gd}{}^{1}$, $T_2{}^{1}$ and $T_{1Gd}{}^{1}$-$T_2{}^{1}$ (numbers of combinations 1, 3 and 6, respectively, in Table 2), whose study region was NCR/NET (indicated by the superscript 1). For combination 1, the value of $d$ was equal to 5; for combination 3, it was equal to 7; and for combination 6, it was equal to 16. Thus, for combination 1, models with one to five variables

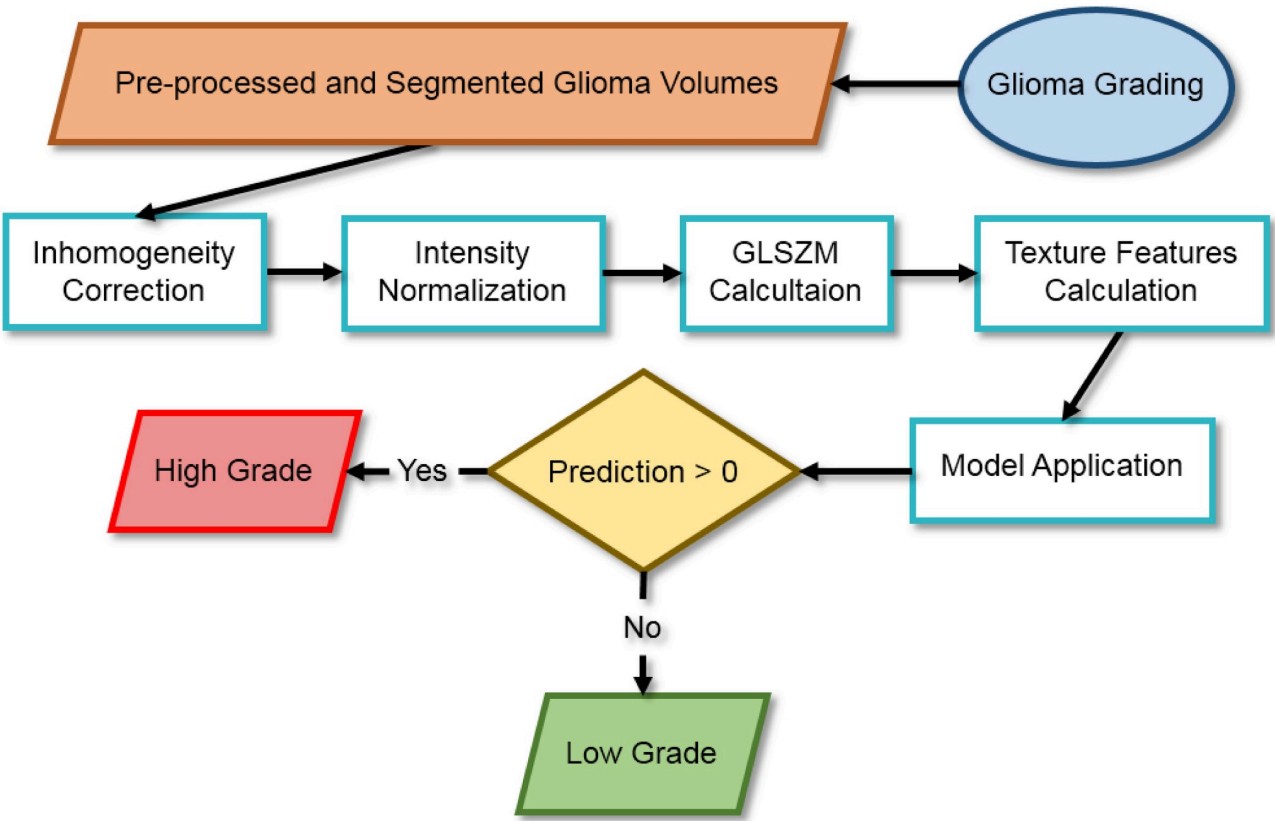

**Fig 3. Flow diagram.** Complete process proposed for the classification of low-grade and high-grade gliomas.

were created; for combination 3, models with one to seven variables were created; and for the combination 6, models with one to sixteen variables were created. Therefore, in total 28 different unique classification models were created (S9 Table).

Fig 4a, 4b and 4c show the results obtained after applying the classification models of combinations 1, 3 and 6 to the testing subset, respectively. The percentages of sensitivity, specificity and accuracy reached are indicated. Fig 4d, 4e and 4f indicate the mae of all the models created. The model that showed the best results (signaled in Fig 4 with black arrow-heads) corresponded to combination 6 ($T_{1Gd}^{1}$-$T_2^{1}$) using the first five ordered highest frequency features. In order, the five features or variables of the models were $F_{szm.z.perc}$, $F_{szm.zs.var}$, $F_{szm.zs.lzlge}$, $F_{szm.zs.lze}$ and $F_{szm.zsnu}$. The first four were measured in $T_2^{1}$ and the fifth one in $T_{1Gd}^{1}$.

Following the methodology for the creation of reduced models, 30 models were created using different combinations among the five variables of the aforementioned best model (S10 Table). Fig 5a shows the sensitivity, specificity and accuracy obtained by the 30 models when they were applied to the testing subset. Their respective mae values are shown in Fig 5b. As it can be seen in Fig 5a, the model that used only the three ordered variables 1-2-5 obtained the same results as the model that used all five variables (1-2-3-4-5). This reduced model obtained a sensitivity of 94.12%, a specificity of 88.24%, an accuracy of 91.18% and a mae of 5.03. Table 3 shows information regarding the three variables (1-2-5) or texture features.

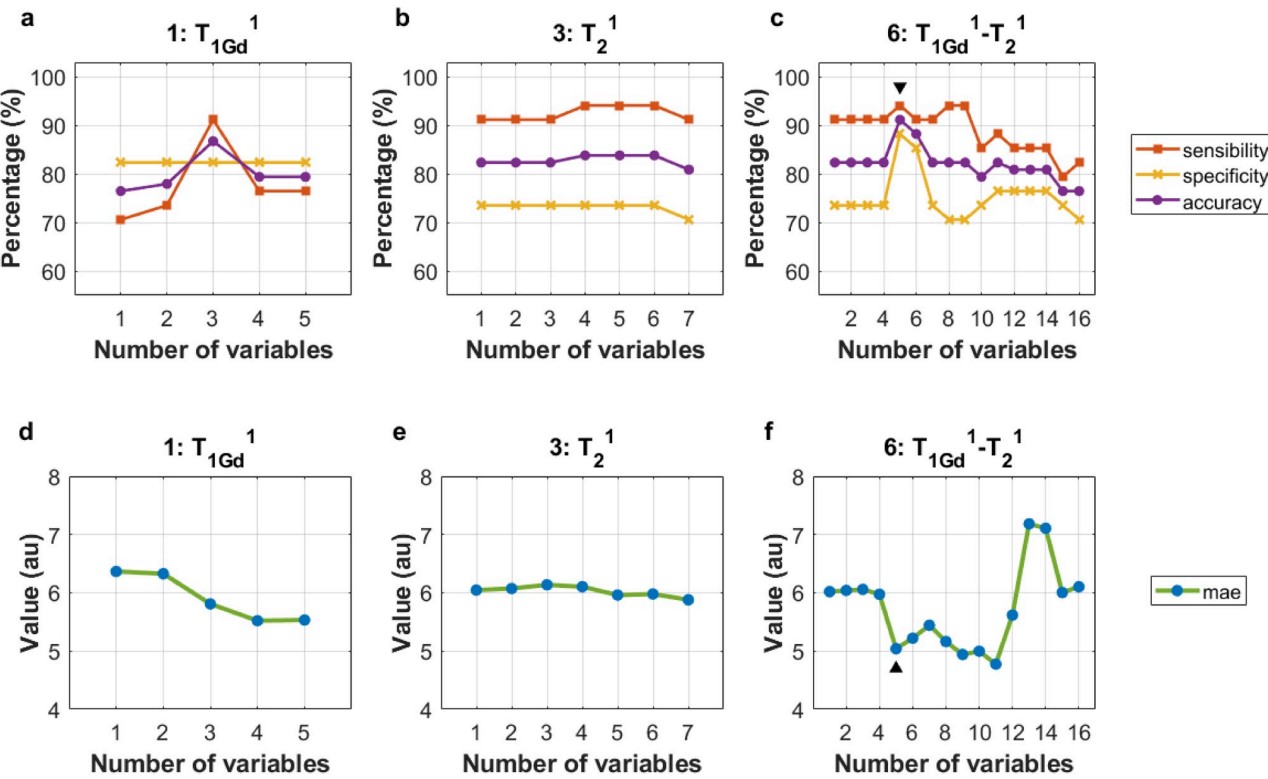

**Fig 4. Results of models.** Three graphs are shown with the results of combinations 1 (a), 3 (b) and 6 (c), using different numbers of variables (horizontal axis). These results consist of the percentages (vertical axis) of sensitivity, specificity and accuracy obtained after applying the models to the testing subset. Three further graphs (d, e and f) indicate the values (in arbitrary units, au) of the mean absolute errors (mae) obtained in each model. The best classification results were obtained in combination 6 by the model with five variables (▼, ▲).

Taking Eq 1 as reference, the mathematical expression of the three-variable reduced model was:

$$\bar{\beta}'_1 x_1 + \bar{\beta}'_2 x_2 + \bar{\beta}'_5 x_5 + \bar{\beta}'_{cte} = \hat{y} \tag{2}$$

and considering the data shown in Table 3, Eq 2 became:

$$13.693 F_{\text{szm.z.perc}} - 0.410 F_{\text{szm.zs.var}} + 31.842 F_{\text{szm.zsnu}} - 19.500 = \hat{y} \tag{3}$$

being the mathematical expression for the best classification model.

In Fig 6, the predictions made by this model when Eq 3 was applied to the testing subset are presented graphically.

In Fig 7, boxplots made from the three variables of the testing LGGs and HGGs are shown. These variables presented significant differences when both study groups were compared ($p = 1.21 \times 10^{-7}$ for $F_{\text{szm.zsnu}}$, and $p = 1.58 \times 10^{-7}$ for $F_{\text{szm.z.perc}}$ and $F_{\text{szm.zs.var}}$). In addition, it can be seen that the testing LGGs had relatively higher values of $F_{\text{szm.zs.var}}$ compared to the testing HGGs, and the testing HGGs had relatively higher values of $F_{\text{szm.zsnu}}$ and $F_{\text{szm.z.perc}}$ compared to the testing LGGs.

After having obtained the best model, the computation time of the complete classification process on the testing subset was measured and averaged. The individual processes carried out were: inhomogeneity correction and intensity normalization of the $T_{1Gd}$ and $T_2$ contrasts;

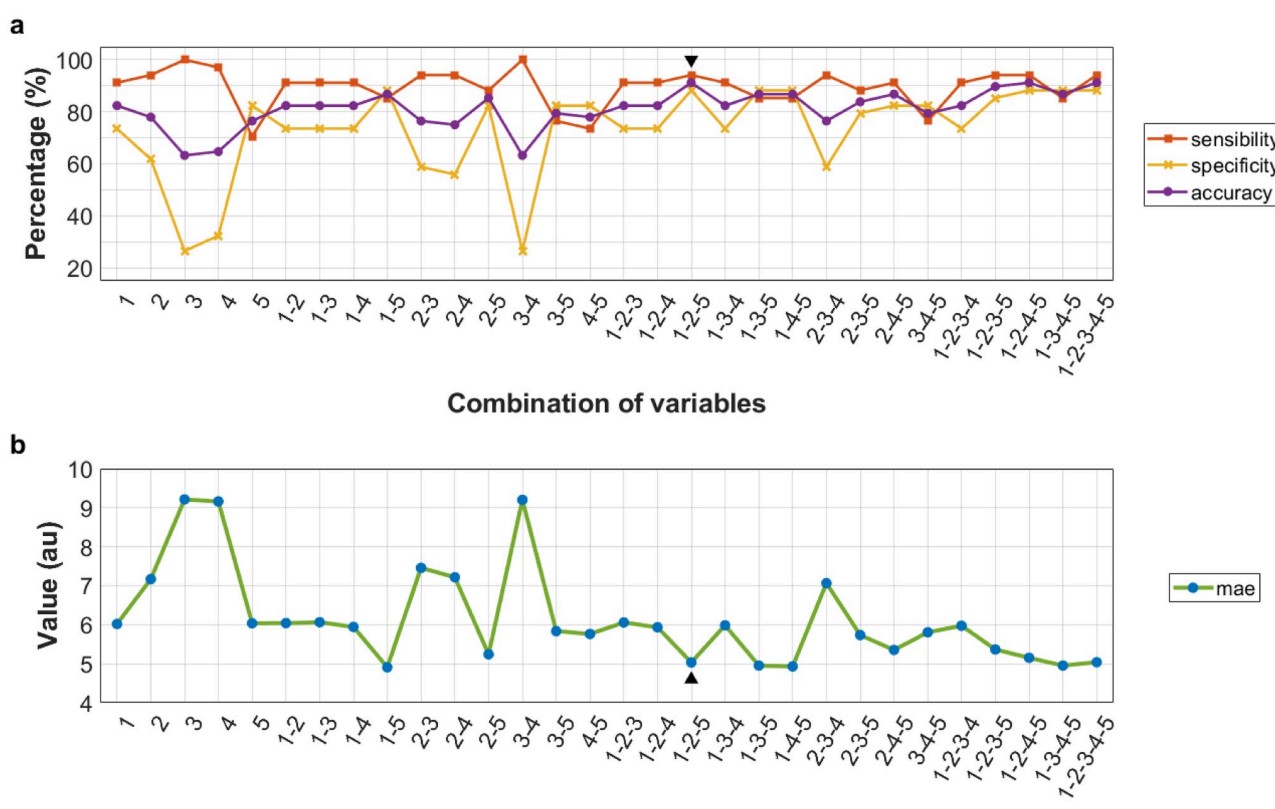

**Fig 5. Results from reduced models.** a. Percentages of sensitivity, specificity and accuracy (vertical axis), obtained by the 30 reduced models, in addition to the combination of variables utilized in each one (horizontal axis), using the following numbering: 1, $F_{szm.z.perc}$; 2, $F_{szm.zs.var}$; 3, $F_{szm.lzlge}$; 4, $F_{szm.lze}$; and 5, $F_{szm.zsnu}$. The first four were measured in $T_2$ contrasts and the fifth in $T_{1Gd}$ contrasts. All features were measured in the NCR/NET region. b. Values (in arbitrary units, au) of the mean absolute errors (mae) obtained in each reduced model. The reduced model that obtained the best results with the lowest number of variables and the smallest error corresponded to the one that combined variables 1-2-5 (▼, ▲).

calculation of two GLSZM, one from $T_{1Gd}$ and another from $T_2$, considering only the NCR/NET region in both; calculation of the three texture features, one being obtained from the GLSZM of $T_{1Gd}$ and two from the GLSZM of $T_2$; application of the model (Eq 3); and classification of gliomas according to the criteria described above. The average time for classification was 2 min 4 s ± 46 s.

**Table 3. Data for the best reduced model.**

| Order ($i$) | Variable ($x_i$) | MRI$^{reg}$ | Coefficient ($\bar{\beta}'_i$) | $p_{min}$ | $p_{max}$ |
|---|---|---|---|---|---|
| 1 | $F_{szm.z.perc}$ | $T_2{}^1$ | 13.693 | $1.47 \times 10^{-7}$ | $8.66 \times 10^{-5}$ |
| 2 | $F_{szm.zs.var}$ | $T_2{}^1$ | -0.410 | $1.47 \times 10^{-7}$ | $8.66 \times 10^{-5}$ |
| 5 | $F_{szm.zsnu}$ | $T_{1Gd}{}^1$ | 31.842 | $1.02 \times 10^{-5}$ | $1.02 \times 10^{-5}$ |
| cte | | | -19.500 | | |

This model used only three variables (1-2-5) obtained from the texture features employed in the five-variable (1-2-3-4-5) model of combination number 6. The order of the variables used is shown, in addition to the MRI contrast and the glioma region from which they were measured. The average coefficient and the minimum and maximum p-values ($p_{min}$ and $p_{max}$, respectively) corresponding to each variable are indicated. The coefficient of the constant term is also shown.

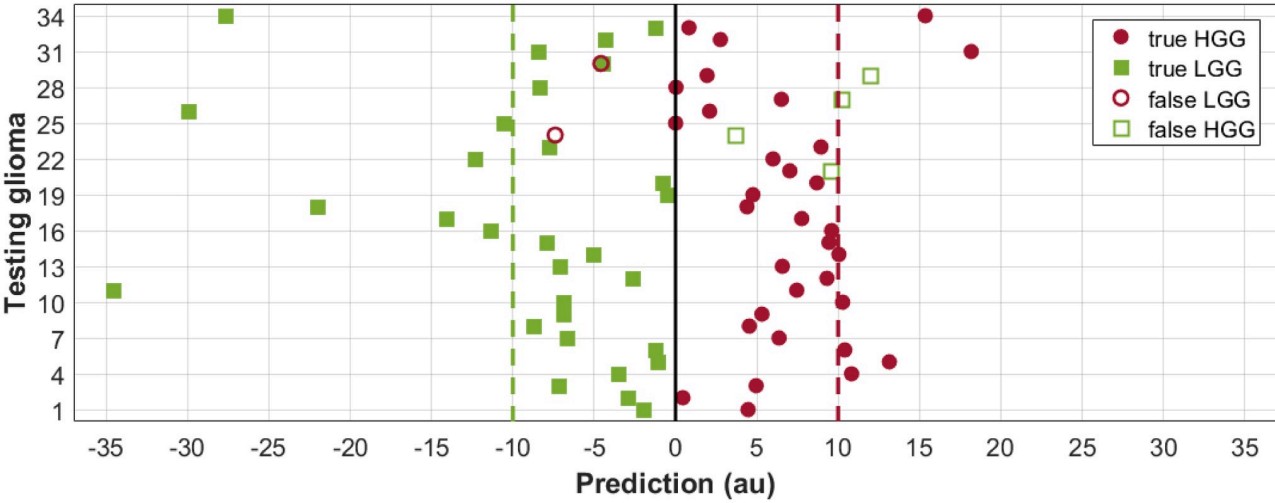

**Fig 6. Predictions made by the best reduced model, when applied to the testing subset.** Testing gliomas (34 LGGs and 34 HGGs; vertical axis) and their predictions (in arbitrary units, au; horizontal axis) are presented. A solid vertical line at zero indicates the chosen threshold. Dotted vertical lines at -10 and 10 indicate the ideal prediction of the LGGs and HGGs, respectively. The filled circles and squares correspond to the true HGGs and true LGGs, respectively, and the empty circles and squares correspond to the false LGGs (or HGGs misclassified) and false HGGs (or LGGs misclassified), respectively.

## Discussion

Through an under-sampling approach to create testing and training subsets with balanced classes, various classification models were created using the highest frequency texture features obtained from the different training subsets. The best model used only three texture features (studying two conventional MRI contrasts and only one glioma region), obtaining good classification results. Thus, this model was characterized by its simplicity, in addition to the reduced average computation time needed to classify an individual glioma. Furthermore, as the

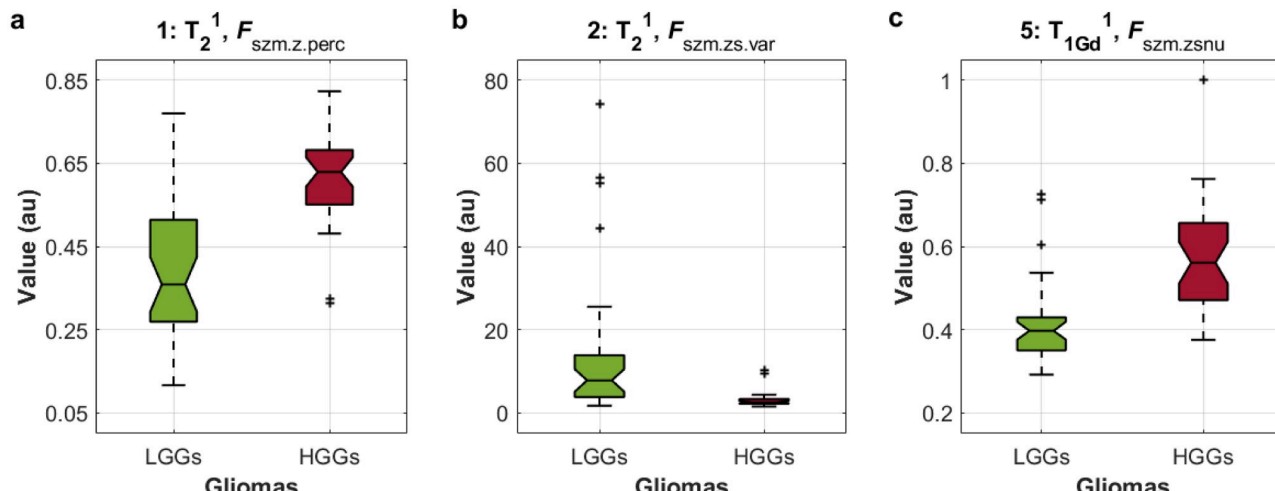

**Fig 7. Boxplots of the texture features or variables 1-2-5, calculated from the testing gliomas.** The grades of the testing gliomas (horizontal axis) and their texture values (in arbitrary units, au; vertical axis) are presented. a. Boxplot of feature number 1, $F_{szm.z.perc}$ (measured in the $T_2^1$ contrast). b. Boxplot of feature number 2, $F_{szm.zs.var}$ (measured in the $T_2^1$ contrast). c. Boxplot of feature number 5, $F_{szm.zsnu}$ (measured in the $T_{1Gd}^1$ contrast).

methodology is thoroughly described and the database studied is publicly available, it is possible to reproduce and corroborate the model reported. Finally, the features used in the model presented significant differences between the testing LGGs and HGGs.

Regarding the interpretation of the variables used in the best model, feature $F_{szm.z.perc}$ (calculated from $T_2^1$) was a measurement of the coarseness of the texture, such that higher values of this feature corresponded to a finer (or more homogeneous) texture [46]. On the other hand, feature $F_{szm.zsnu}$ (calculated from $T_2^1$) measured the variability of zone size volumes across the image, with a higher value indicating more heterogeneity in zone size volumes [46], while feature $F_{szm.zs.var}$ (calculated from $T_{1Gd}^1$) measured the variance of the zone sizes, with higher values indicating a more heterogeneous texture [27]. From the interpretation of the features and the results described above, it could be deduced that LGGs had a more heterogeneous texture than HGGs, specifically in the $T_2$ contrasts; and HGGs had a more heterogeneous texture than LGGs, specifically in the $T_{1Gd}$ contrasts, with both cases studying the NCR/NET region. Several works have reported models whose main classification variable was heterogeneity of gliomas [18, 23, 25, 47–49]. For example, through texture analysis applied on diffusion tensor imaging [25, 49] and diffusion kurtosis imaging [49] maps, diverse features that characterized the heterogeneity of gliomas indicated an increased heterogeneity for higher-grade gliomas compared to lower-grade gliomas. Moreover, Kin et al. [47] studied the texture matrix called the grey level co-occurrence matrix (GLCM) on contrast-enhanced T1 MR and ADC maps, and reported higher values of entropy (or non-uniformity) as well as reduced values of homogeneity for HGGs, when these were compared to LGGs. Also, Skogen et al. [48] applied texture analysis on post-contrast spoiled gradient echo (SPGR) sequences using a filtration histogram technique in order to obtain features from fine to coarse, and quantified the heterogeneity of gliomas through the standard deviations of the histograms. They reported results that showed a higher heterogeneity for the HGGs compared to the LGGs. Hence, diverse studies have related higher heterogeneity to higher-grade glioma. However, the present work showed that one glioma grade had a more heterogeneous texture than the other, depending on the studied MRI contrast. Therefore, this result is complementary to what is usually reported, since it was more specific after having included the MRI contrast as a variable in the models.

One of the objectives of this work was to present one classification model explicitly, and then apply it on a single and independent testing subset as a validation process. Because of this, the database was divided between different training subsets and one testing subset, creating the models from the first and applying them to the last one. The number of 30 LGGs and 30 HGGs was chosen to form the training subsets, because 30 was the minimum number of gliomas per study group such that there were no significant differences in the results obtained by the created models (data not reported). In addition, the same number of LGGs and HGGs were chosen to avoid the problem of so-called "class imbalance" using an under-sampling approach. Later, as part of this approach, complementary information obtained from different training subsets was used to create the classification models.

The main contribution of this work, in addition to the proposed under-sampling approach already mentioned, is the simplicity of the best classification model (which obtained high values of accuracy) compared to others recently reported. For instance, Wang et al. [5] analyzed a combination of advanced and conventional MRI (diffusion-weighted, contrast-enhanced T1-weighted and axial T2-weighted images) of 26 LGGs and 26 HGGs divided into a training and validation set. A total of 654 radiomic features were extracted for each subject. Through a LASSO regression 15 features were chosen, from which a nomogram was created. Then, the classification capacity of the nomogram was evaluated using the Harrell's concordance index (C-index), obtaining a C-index of 0.971 and 0.961 on training and validation data, respectively.

In another work, Khawaldeh et al. [7] studied 2D slide images from conventional MRI (FLAIR) of 128 subjects including LGGs, HGGs and healthy subjects. The authors proposed a modified version of the convolutional neural network known as AlexNet [50] in 2D, which reached an accuracy of 91.16% by differentiating the three study groups. On the other hand, Tian et al. [24] analyzed conventional (T1-weighted images before and after contrast-enhancement, and T2-weighted images) and advanced (multi-b-value diffusion-weighted and 3D arterial spin labeling images) MRI of 42 LGGs and 111 HGGs by extracting texture features and histogram parameters. SVM-based recursive feature elimination was used to choose the best features for the classification of gliomas, and then create different SVM classifiers by cross-validation. Using 30 texture features they reached an average classification accuracy of 96.8%. Also, Gupta et al. [20] analyzed conventional MRI (T1-weighted images before and after contrast-enhancement, T2-weighted and FLAIR images) of 80 LGGs and 120 HGGs to perform three tasks: detection, location and identification of gliomas. For the third task (identification), they used geometric parameters such as area, solidity, perimeter and orientation of the tumor, in addition to consultations with radiologists. They obtained an accuracy for classifying LGGs and HGGs of 94.4% and 94%, respectively, when T1-weighted images before and after contrast-enhancement were studied, and 96.5% and 97% when they studied T2-weighted and FLAIR images. Therefore, in this work conventional MRI ($T_{1Gd}$ and $T_2$ contrasts) was studied, while others have analyzed advanced MRI or a combination of both [5, 21–24, 51–54]. The model was created from a simple mathematical method (a multiple linear regression), in comparison to others in which mathematical tools of higher complexity were utilized [7, 52–54]. The best model was found to use only three variables of a single type (quantitative, being also only texture features), instead of a combination of different classes and types of variables [21, 24, 51, 53]. A texture analysis was performed (which is easy to implement for any type of MRI) and a single texture matrix was used instead of different matrices [24], with the chosen one (GLSZM) being a suitable texture matrix when heterogeneity is a predominant characteristic of the object of study. In addition, since the studied database is publicly available and the mathematical expression of the best model has been explicitly reported, the reproducibility of the methodology presented and the corroboration of the results by other independent studies is feasible. In general, any classifier model has a very strong dependence on the database and image acquisition protocol used to develop them. Usually an institutional database and protocol are used for this purpose. In contrast, the BRATS database was obtained from 19 study centers with different clinical protocols and various scanners. This makes the database heterogeneous and therefore it better approximates a realistic scenario of what could be found in a clinical environment. Hence, there is a possibility that the reported model could be tested on other databases without being limited to a specific clinical protocol. In addition to the simplicity of the reported classification model, since conventional MRI and texture analysis were studied, the diagnostic model presented is low-cost and easy-to-implement, so that it is accessible to populations with reduced economic and scientific resources.

Among the limitations of the work presented, the following should be mentioned. Since there was only a single independent testing subset (randomly chosen), there is a possibility that the results may vary according to the chosen subset. Also, the number of gliomas that made up the training and testing subsets were relatively small. It is always preferable and desirable to have a database with a greater number of samples, such that the results obtained have a higher reliability. On the other hand, images of manually segmented gliomas were used, so that the proposed classification method was supervised (not fully automated). Moreover, the criterion for the choice of the texture features was limited to use only statistical tests. This was not enough to ensure good results in all models, even though the texture features used in them showed significant differences. Moreover, the molecular characteristics of the tumors have

shown to be more useful than the histological characteristics in the diagnosis, treatment and prognosis of the patients. Taking all of this into account, future work should consider applying the reported classification methodology to other independent databases. An automatic segmentation method must be developed or an existing one must be implemented, such that the glioma classification methodology becomes fully automated. Besides, other criteria for the extraction (principal component analysis, linear discriminant analysis, etc.) and selection (filter approach, wrapper approach, etc.) of texture features should be considered. Also, other texture matrices (gray level co-occurrence matrix, grey level run length matrix, etc.) and conventional MRI contrasts (T1, FLAIR, etc.) could be studied. Finally, the work performed here, and the characteristics studied, are intended to complement other analysis techniques, such as those that study molecular characteristics, so that future work can include correlation and combination of results from different approaches.

In conclusion, the methodology proposed has proven to be useful for the classification of low-grade and high-grade gliomas, obtaining high levels of accuracy. The main objective of the authors is that the model can be implemented as a complementary technique in the clinical diagnosis environment for this type of brain tumors.

## Supporting information

**S1 Table. List of HGGs and LGGs, including the key names used in the database and the reference numbering used in the work.**
(DOCX)

**S2 Table. Reference gliomas for normalization.**
(DOCX)

**S3 Table. Testing glioma subset.**
(DOCX)

**S4 Table. One hundred training glioma subsets.**
(DOCX)

**S5 Table. Texture features of all gliomas, both training and testing, for the MRI$^{reg}$ T$_{1Gd}$$^1$.**
(DOCX)

**S6 Table. Texture features of all gliomas, both training and testing, for the MRI$^{reg}$ T$_2$$^1$.**
(DOCX)

**S7 Table. Texture features of all gliomas, both training and testing, for the MRI$^{reg}$ T$_{1Gd}$$^2$.**
(DOCX)

**S8 Table. Texture features of all gliomas, both training and testing, for the MRI$^{reg}$ T$_2$$^2$.**
(DOCX)

**S9 Table. Models created for combinations 1, 3 and 6, indicating the respective ordered features and their coefficients.**
(DOCX)

**S10 Table. Thirty reduced models created from the first five ordered highest frequency features used in combination 6, indicating order, reference numbering and their coefficients.**
(DOCX)

## Author Contributions

**Conceptualization:** José Gerardo Suárez-García.

**Data curation:** José Gerardo Suárez-García.

**Formal analysis:** José Gerardo Suárez-García.

**Funding acquisition:** José Gerardo Suárez-García, Benito de Celis-Alonso.

**Investigation:** José Gerardo Suárez-García.

**Methodology:** José Gerardo Suárez-García.

**Project administration:** José Gerardo Suárez-García, Benito de Celis-Alonso.

**Resources:** José Gerardo Suárez-García.

**Software:** José Gerardo Suárez-García.

**Supervision:** José Gerardo Suárez-García, Javier Miguel Hernández-López, Eduardo Moreno-Barbosa, Benito de Celis-Alonso.

**Validation:** José Gerardo Suárez-García.

**Visualization:** José Gerardo Suárez-García.

**Writing – original draft:** José Gerardo Suárez-García, Benito de Celis-Alonso.

**Writing – review & editing:** José Gerardo Suárez-García, Benito de Celis-Alonso.

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
