## [Decision Letter · Decision Letter 0]

9 Apr 2020

PONE-D-20-01908

A simple model for glioma grading based on texture analysis applied to conventional brain MRI

PLOS ONE

Dear Dr. de Celis-Alonso,

Thank you for submitting your manuscript to PLOS ONE. After careful consideration, we feel that it has merit but does not fully meet PLOS ONE’s publication criteria as it currently stands. Therefore, we invite you to submit a revised version of the manuscript that addresses the points raised during the review process.

We would appreciate receiving your revised manuscript by May 24 2020 11:59PM. To enhance the reproducibility of your results, we recommend that if applicable you deposit your laboratory protocols in protocols.io, where a protocol can be assigned its own identifier (DOI) such that it can be cited independently in the future. For instructions see: http://journals.plos.org/plosone/s/submission-guidelines#loc-laboratory-protocols

We look forward to receiving your revised manuscript.

Kind regards,

Jonathan H Sherman

Academic Editor

PLOS ONE

Journal Requirements:

Reviewers' comments:

Reviewer's Responses to Questions

**Comments to the Author**

1. Is the manuscript technically sound, and do the data support the conclusions?

Reviewer #1: Yes

Reviewer #2: Yes

2. Has the statistical analysis been performed appropriately and rigorously? 

Reviewer #1: Yes

Reviewer #2: Yes

3. Have the authors made all data underlying the findings in their manuscript fully available?

Reviewer #1: Yes

Reviewer #2: Yes

4. Is the manuscript presented in an intelligible fashion and written in standard English?

Reviewer #1: Yes

Reviewer #2: Yes

5. Review Comments to the Author

Reviewer #1: The authors present study on texture analysis for differentiating low and high-grade gliomas. Although there are many similar studies on texture analyses for gliomas, the approach presented here is unique. In particular, they used an external database, suggesting the wide applicability of their work. The work can benefit from minor grammatical editing.

Reviewer #2: Thankful the authors for the exelent work. I don't have any specific issue to address here. Very inteligent design and analyis, clear stated question and good sientific english style.

Highly recommended for the pablication.

6. PLOS authors have the option to publish the peer review history of their article (what does this mean?). If published, this will include your full peer review and any attached files.

Reviewer #1: No

Reviewer #2: Yes: Gayane Aghakhanyan, MD, PhD

---

## [Author Response · Author response to Decision Letter 0]

14 Apr 2020

Dear Editor and reviewers:

We thank you for your comments and considerations. In general reviews were really positive and enthusiastic and no major criticism to methodology, impact and results was raised. In fact, we were congratulated by reviewers which is not that common. We would like to thank them both for this. 

We have addressed the minor points raised by reviewers and editor in the following manner: 

1. The main criticism of reviewer 1, was exclusively: “The work can benefit from minor grammatical editing”. We have sent the manuscript to a native speaking editing company (see file: certificate.pdf attached in this resubmission). This has greatly improved the work, and we believe the paper now has very high standards in this aspect. 

2. Editor asked to adapt the manuscript to PLOS ONE style format. We have done so thoroughly. Images were passed through website which checked image format and quality and passed but were re-dimensioned. Because of this in the resubmission, we attach all images and supplementary material again. 

3. No laboratory specific lab protocols were used for this publication, so we did not consider the petition to load them to the journals site applicable. Therefore, no specific protocol from this work was submitted to the journal. 

Please find enclosed in this re-submission:

• Revised manuscript with track changes (pdf and .tex format). This is manuscript with highlighted changes in it (yellow). 

• Manuscript (pdf and .tex format). This is manuscript with changes already implemented in it. 

• Certificate (pdf). Certificate of style and English formatting. 

We hope all these changes are enough to get this work accepted in your journal. 

Yours,

Dr. Benito de Celis Alonso

---

## [Decision Letter · Decision Letter 1]

30 Apr 2020

A simple model for glioma grading based on texture analysis applied to conventional brain MRI

PONE-D-20-01908R1

Dear Dr. de Celis-Alonso,

We are pleased to inform you that your manuscript has been judged scientifically suitable for publication and will be formally accepted for publication once it complies with all outstanding technical requirements.

With kind regards,

Jonathan H Sherman

Academic Editor

PLOS ONE

Additional Editor Comments (optional):

Reviewers' comments:

Reviewer's Responses to Questions

**Comments to the Author**

1. If the authors have adequately addressed your comments raised in a previous round of review and you feel that this manuscript is now acceptable for publication, you may indicate that here to bypass the “Comments to the Author” section, enter your conflict of interest statement in the “Confidential to Editor” section, and submit your "Accept" recommendation.

Reviewer #1: All comments have been addressed

Reviewer #2: All comments have been addressed

2. Is the manuscript technically sound, and do the data support the conclusions?

Reviewer #1: Yes

Reviewer #2: (No Response)

3. Has the statistical analysis been performed appropriately and rigorously? 

Reviewer #1: Yes

Reviewer #2: Yes

4. Have the authors made all data underlying the findings in their manuscript fully available?

Reviewer #1: Yes

Reviewer #2: Yes

5. Is the manuscript presented in an intelligible fashion and written in standard English?

Reviewer #1: Yes

Reviewer #2: Yes

6. Review Comments to the Author

Reviewer #1: Thank you for completing the suggested English language editing. This will help with the wide dissemination of your important work.

Reviewer #2: All queries were address. No additional comments on behalf of me. I'd recommend the manuscript for publication.

7. PLOS authors have the option to publish the peer review history of their article (what does this mean?). If published, this will include your full peer review and any attached files.

Reviewer #1: No

Reviewer #2: Yes: Gayane Aghakhanyan, MD, PhD

---

## [Editor Report · Acceptance letter]

4 May 2020

PONE-D-20-01908R1 

A simple model for glioma grading based on texture analysis applied to conventional brain MRI 

Dear Dr. de Celis-Alonso:

I am pleased to inform you that your manuscript has been deemed suitable for publication in PLOS ONE. Congratulations! Your manuscript is now with our production department. 

With kind regards,

on behalf of

Dr. Jonathan H Sherman 

Academic Editor

PLOS ONE